# Sub-nanowatt microfluidic single-cell calorimetry

Sahngki Hong[1,2], Edward Dechaumphai[2,5], Courtney R. Green [3,5], Ratneshwar Lal [1,2,3], Anne N. Murphy[4], Christian M. Metallo[3] & Renkun Chen [1,2✉]

Non-invasive and label-free calorimetry could become a disruptive technique to study single cell metabolic heat production without altering the cell behavior, but it is currently limited by insufficient sensitivity. Here, we demonstrate microfluidic single-cell calorimetry with 0.2-nW sensitivity, representing more than ten-fold enhancement over previous record, which is enabled by (i) a low-noise thermometry platform with ultralow long-term (10-h) temperature noise (80 µK) and (ii) a microfluidic channel-in-vacuum design allowing cell flow and nutrient delivery while maintaining a low thermal conductance of $2.5\,\mu\text{W}\,\text{K}^{-1}$. Using *Tetrahymena thermophila* as an example, we demonstrate on-chip single-cell calorimetry measurement with metabolic heat rates ranging from 1 to 4 nW, which are found to correlate well with the cell size. Finally, we perform real-time monitoring of metabolic rate stimulation by introducing a mitochondrial uncoupling agent to the microchannel, enabling determination of the spare respiratory capacity of the cells.

[1] Materials Science and Engineering Program, University of California San Diego, La Jolla, CA 92093, USA. [2] Department of Mechanical and Aerospace Engineering, University of California San Diego, La Jolla, CA 92093, USA. [3] Department of Bioengineering, University of California San Diego, La Jolla, CA 92093, USA. [4] Department of Pharmacology, University of California San Diego, La Jolla, CA 92093, USA. [5]These authors contributed equally: Edward Dechaumphai, Courtney R. Green. ✉email: rkchen@ucsd.edu

Many biological processes are inherently tied to heat generation or absorption, which can be measured calorimetrically[1]. The metabolic activities of microorganisms also involve heat generation as a product of adenosine triphosphate (ATP) synthesis and oxidative phosphorylation[2]. Calorimetry is a powerful tool to investigate the metabolic activities because it is a noninvasive (i.e., without the need of cell lysis to extract and analyze the metabolites) and label-free technique, enabling monitoring of the metabolic behavior of living cells in real time without altering the nature of the microorganisms[3]. Calorimetry can also be applied to diagnose bacterial disease by detecting the metabolic heat of bacteria in the blood of a patient and to detect antibiotic resistance[4]. Abnormal metabolic activities of cancer cells can also be probed by calorimetry, and a large amount of research has been devoted to elucidating the relationship between the metabolic heat of cells and cancer[5]. Metabolic heat has also been recently measured to reveal the energetics of embryonic cell development[6].

Currently, the metabolic heat of cells is measured by averaging across the entire population of numerous cells. However, the average cell property does not necessarily represent the property of individual cells[7] because the cells are highly heterogeneous in terms of the type and amount of proteins[8], respiration rate[9], size and mechanical properties[10], and division time[11]. In particular, heterogeneity of mitochondria such as diverse membrane potentials, redox states, and calcium levels leads to various metabolic behaviors of individual cells[12]. Calorimetry has a unique advantage for the investigation of these metabolic heterogeneities compared with other methods; for instance, it can provide the overall metabolic information including that for anaerobic metabolism, which cannot be detected by oxygen consumption rate analysis. Although these heterogeneities have been widely investigated using optical[13], chemical[14], electrical[15], and mechanical[10] cytometry techniques, a method that can measure single-cell heat production rate has yet to be developed. This is mainly because the sensitivity of state-of-the-art calorimeters is insufficient to measure the metabolic heat from a single cell. The poor sensitivity is attributed to the inherent difficulty in measuring heat; various parasitic heat loss mechanisms such as conduction, convection, and radiation easily dissipate the small amount of heat produced from a single cell, which is usually less than a few nW.

Hence, efforts to enhance the calorimetry sensitivity ($Q_s$) have been centered on minimizing the thermal conductance ($G$) to suppress the parasitic heat loss as well as reducing the minimum detectable temperature ($\Delta T_{min}$) as $Q_s = G \times \Delta T_{min}$. In particular, the introduction of microfabrication technology to calorimetry enabled the integration of high-resolution thermometers and a microfluidic cell-delivery system on a thermally insulative thin membrane, resulting in significant improvement of the calorimeter sensitivity[16–28]. Microfluidic calorimeters have also allowed dosing stimuli onto microorganism or monitoring of theomorphological change and population of cells in real time without disturbance of the measurement[26,27].

Microfluidic calorimeters have evolved over the last few decades to measure metabolic heat production from microorganisms. Johannessen et al.[26] demonstrated a chip calorimeter possessing a sensitivity of 13 nW. The calorimeter was able to measure 16.3 nW of metabolic heat from ten brown adipocytes stimulated with noradrenaline. A major breakthrough occurred in 2009, when Lee et al.[16] developed a chip calorimeter with a sensitivity of 4.2 nW. They achieved low thermal conductance by virtue of thin parylene membrane housed in a vacuum environment. However, the device was not applied for cell measurements, presumably because the detection limit remained higher than heat production rate by single cells, even in cells with high metabolic rate such as

*Tetrahymena* (~ 3 nW)[29–31]. For about a decade since the pioneering work of Lee et al.[16], no calorimeter has demonstrated sensitivity better than 4 nW, highlighting the challenges of improving the sensitivity of chip calorimetry with microfluidic handling capability.

Here, we present a chip calorimeter capable of single-cell metabolic heat measurement with a high sensitivity of 0.2 nW. We achieve approximately an order of magnitude greater sensitivity by implementing a one-dimensional suspended microfluidic design in vacuum and a measurement platform with long-term stable temperature (80 μK temperature drift in 10 h). Furthermore, we achieve single-cell metabolic measurement by magnetically trapping the cells in the microfluidic channel for reliable thermal measurement without perturbation introduced by cell movement. The microfluidic platform and the trapping technique also allow for a continuous supply of the fluid containing nutrients and oxygen to the cells. The high sensitivity and accurate cell control system enable us to measure the nW level of heat production from single *Tetrahymena thermophile*. Our measurement reveals a positive correlation between the metabolic heat generation and cell size. Finally, we demonstrate real-time monitoring of metabolic heat increase induced by the injection of the mitochondrial uncoupling agent carbonyl cyanide 4-(trifluoromethoxy)phenylhydrazone (FCCP) through the microfluidic channel, which is used to extract the spare respiratory capacity (SRC). Given the tremendous potential of probing the metabolic rate at the individual cell level, our calorimetry platform will open up a pathway for a wide range of fundamental biological studies and potentially biomedical applications.

## Results

**Design of calorimeter with high sensitivity.** The development of a high-sensitivity calorimeter relies on minimization of the thermal conductance and minimum detectable temperature. Previous two-dimensional membrane designs still have large thermal conductance. In our design, we utilized a one-dimensional (1D) suspended microfluidic channel with a small cross-sectional area and large length ($2L = 5.0$ mm) to achieve low conduction and radiation thermal conductance (Fig. 1a). The microfluidic channel is composed of parylene, which has high mechanical strength, good visual transparency, and, most importantly, low thermal conductivity (Supplementary Fig. 2b) compared with other materials widely used for chip calorimeters (e.g., $Si_3N_4$[22,24,26], glass[25], Si[27]). The parylene backbone layer had a width of 120 μm and thickness of 12 μm and formed the microfluidic tube (Fig. 1b). The microfluidic channel had a cross-sectional area of $35 \times 50$ μm$^2$, which is compatible with most cell sizes (<35 μm in diameter). The entire tube structure was suspended on a silicon wafer with a ~ 200-μm gap. Lastly, the entire device was placed in a high-vacuum environment (<0.1 mT)[16] to eliminate convection heat loss (Fig. 1d, e). The fluids into and out of the fluidic channel were introduced via vacuum feedthroughs. Photographs of the setup are shown in Supplementary Fig. 6.

A Bi–Pt thermopile and Pt heater were microfabricated and integrated in the parylene layer underneath the microfluidic channel (Fig. 1c). The thermopile measured the temperature difference ($\Delta T$) between the measuring junctions ($T_{mea}$) in the middle of the channel and the reference junctions ($T_{ref}$) on the Si wafer (Fig. 1a). The Pt heater was used to characterize the thermal conductance of the fluidic channel by applying joule heating to the channel. The thermopile was calibrated by raising the temperature of the junction to known temperatures with the Pt heater (Fig. 1c) and measuring the voltage of the thermopile. The thermopile was designed by considering the tradeoff between the thermal conductance and temperature sensitivity: increasing

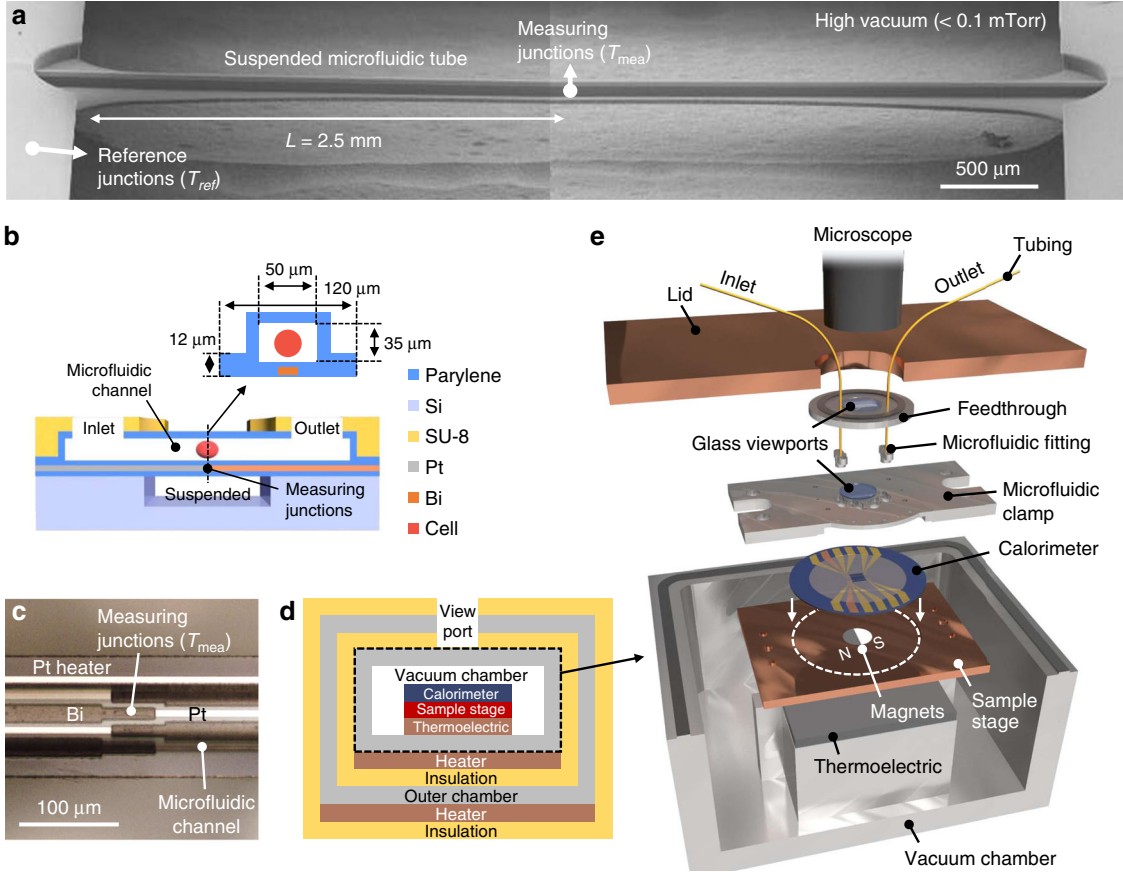

**Fig. 1 Design of single-cell calorimeter and measurement platform. a** SEM image of calorimeter with a 5-mm long (2L) one-dimensional suspended microfluidic channel. The calorimeter is surrounded by a vacuum environment to reduce heat loss. More than 10 samples (n > 10) with the same geometry were reproducibly fabricated. **b** Cross-sectional schematic illustration of calorimeter. **c** Photograph of middle of suspended channel, including measuring junctions of the Bi–Pt thermopile and a Pt heater. **d** Schematic illustration of thermal insulation design of measurement platform with three temperature-controlled layers (outer chamber, vacuum chamber, and sample stage). **e** Schematic illustration of components of the vacuum chamber. Photographs of the setup are shown in Supplementary Fig. 6.

the number of thermopile junctions leads to better temperature detection sensitivity but also increases the thermal conductance. Likewise, reducing the resistance of the thermopile decreases the electrical noise (e.g., Johnson or 1/f noise) but results in higher thermal conductance. Our optimized thermopile was composed of four pairs of Bi and Pt thin films (Fig. 1c and Supplementary Fig. 1b), and the root-mean-square (rms) voltage noise of the measurement system was 19 nV, which includes noises from the thermopile, an operational amplifier (CS3002, Cirrus Logic), and a low-pass filter (cutoff frequency: 0.016 Hz), as shown in Supplementary Fig. 4a.

The overall thermal conductance of our calorimeter including the aforementioned components and water in the microfluidic channel was estimated as a function of the half channel length (L) using a fin equation (see Supplementary Note 1 for derivation):

$$G = 2mkA_c\left(\frac{e^{mL} + e^{-mL}}{e^{mL} - e^{-mL}}\right) \quad (1)$$

where the fin parameter $m = \sqrt{\frac{hP}{kA_c}}$, k is the thermal conductivity, $A_c$ is the cross-sectional area of the microfluidic channel, P is the outer perimeter of the microfluidic channel, and h is the radiation heat-transfer coefficient ($h = 4\varepsilon\sigma T_{avg}^3$, where $\sigma$ is the Stefan–Boltzmann constant, $\varepsilon$ is the emissivity, and $T_{avg}$ is the average temperature). Equation (1) shows that the overall thermal conductance is saturated at L = 2.5 mm, where radiation heat loss

becomes more important than the conduction heat loss because of the large surface area (Supplementary Fig. 3a). The estimated overall thermal conductance with L of 2.5 mm is expected to be 2.48 μW K$^{-1}$, including the backbone parylene layers, water inside the microfluidic channel, and Bi–Pt thermopile. It is worth noting that ~48% of the total thermal conductance comes from the water channel (Supplementary Fig. 3b), which is needed for the continuous nutrient and oxygen supply. We also optimized the geometry to ensure the temperature uniformity around the cell (Supplementary Fig. 3c) and mechanical integrity of the channel–substrate junction (Supplementary Fig. 3d).

The fabricated device was loaded onto our measurement platform (Fig. 1d, e), which was designed to minimize the baseline temperature drift and provide a vacuum environment. The temperature stability is especially important in single-cell calorimetry because external agitations such as light illumination from a microscope used to visualize the cell in real time and fluid flow are inevitable. We minimized the temperature drift by using three levels of thermal insulation and temperature-control layers (Fig. 1d) as well as a stable and hermetically sealed fluid control system (Fig. 1e). By implementing these extensive thermal and fluidic control schemes, we were able to achieve a baseline temperature stability of our calorimeter of within 80 μK for more than 10 h (Fig. 2a) under the condition of microscope illumination. We also showed that the thermal conductance and temperature stability were similar when the fluid in the

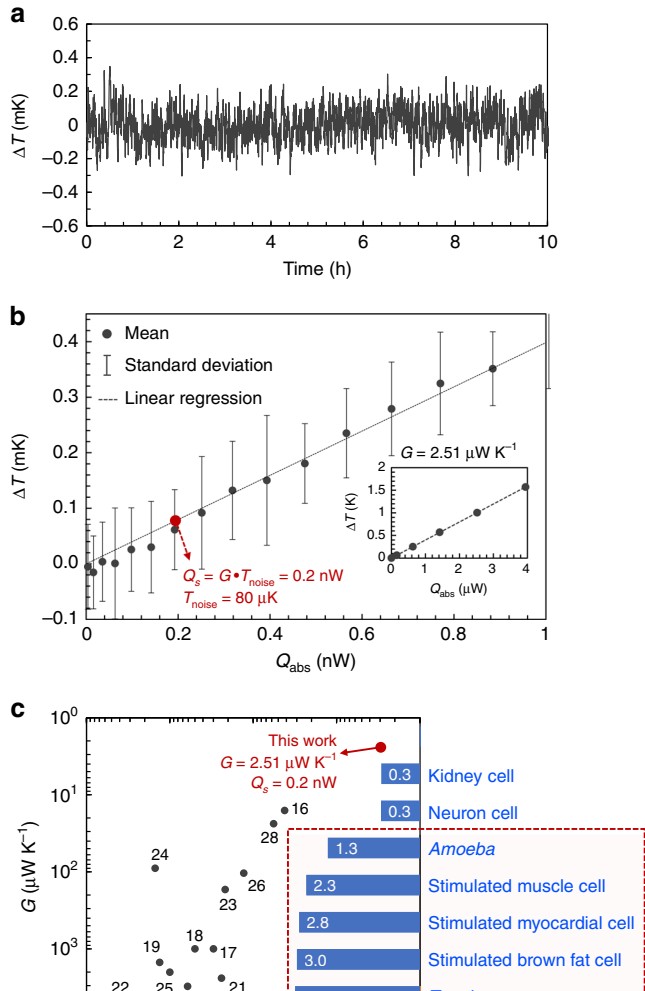

**Fig. 2 Baseline temperature stability and sensitivity of calorimeter.**
**a** Temperature fluctuation of microfluidic channel measured by Pt/Bi thermopile for 10 h under microscope illumination. Similar results with 0.1 mm s$^{-1}$ fluid flow are shown in Supplementary Fig. 7a, c. **b** Temperature rise of calorimeter in response to Joule heating from embedded Pt heater. The mean $\Delta T$ and rms temperature noise ($T_{noise}$) based on the temperature measurement results collected every 0.5 s for 5 min ($n = 600$), are displayed as a function of absorbed power ($Q_{abs}$). $T_{noise}$ was 80 μK and the thermal conductance of the calorimeter was 2.51 μW K$^{-1}$ ($G = dQ_{abs}/d\Delta T$, extracted from the same plot with a wider range of $Q_{abs}$ shown in the inset) corresponding to calorimetry sensitivity of 0.2 nW ($Q_s = G \cdot T_{noise}$). **c** Comparison of $G$ (y-axis) and $Q_s$ (x-axis) of previous calorimeters[16–28] and ours in this work. The numbers next to the filled circles are the reference numbers of the papers reporting the corresponding calorimeters. The blue horizontal bars show the expected metabolic rates of typical cells. Source data for 2**a, b** are provided as a Source Data file.

microchannel was flowing at a speed of 0.1 mm s$^{-1}$ (Supplementary Fig. 7a, c), which was needed to provide sufficient nutrient and oxygen supply to single *Tetrahymena* cells (see next section).

The sensitivity of our calorimeter was demonstrated by heating the suspended channel with the embedded Pt heater and monitoring the voltage from the Bi–Pt thermopile. The heating power absorbed by the measuring junctions ($Q_{abs}$) gradually increased from 0 to 4 μW, and the voltage output from the thermopile was collected for 5 min at each absorbed power. The

voltage output from the thermopile was converted to a temperature signal ($\Delta T$) using the Seebeck coefficient ($S = 229$ μV K$^{-1}$), and the average $\Delta T$ and rms temperature noise ($T_{noise}$) of each measurement are plotted in Fig. 2b. The measured thermal conductance of our calorimeter was 2.51 μW K$^{-1}$ ($G = dQ_{abs}/d\Delta T$), which is consistent with the analytical estimation obtained using Eq. (1) (2.48 μW K$^{-1}$). This thermal conductance is approximately six times lower than that of the state-of-the-art chip calorimeter (16 μW K$^{-1}$)[16]. $T_{noise}$ was 80 μK for 5 min, and the corresponding sensitivity of our single-cell calorimeter was 0.2 nW ($Q_s = G \times T_{noise}$). This sensitivity is approximately one order of magnitude better than that of state-of-the-art on-chip calorimeters (4.2–13 nW)[16,26], as summarized in Fig. 2c. The expected metabolic rates of several cells also shown in Fig. 2c; it is apparent that our calorimetry platform achieved sufficient sensitivity to measure the metabolic rate of single cells such as protozoa[29–31] or stimulated mammalian cells[32], which have not been able to be measured by previous calorimeters[16–28].

**Single-cell trapping and continuous nutrient supply.** As a demonstration of our calorimeter and measurement platform, we measured metabolic heat generation from single *Tetrahymena thermophila* SB210. *Tetrahymena* is widely used as a model cell line to study mammalian cells because it has high homology with mammalian cells not only responding to mammalian hormones but also synthesizing similar molecules (e.g., insulin)[33]. Therefore, their metabolism has been studied by monitoring their oxygen consumption[34–37] or metabolic heat[29–31] or the size of their mitochondria;[38] however, none of the existing measurement platforms have enabled single-cell metabolic rate analysis. Using our high-sensitivity single-cell calorimeter, we were able to measure the metabolic rate of single *Tetrahymena* with sub-nW resolution.

A key challenge to measure *Tetrahymena* (and other cells) is the need to trap individual cells in the microfluidic channel and supply sufficient nutrients and oxygen to the trapped cells. Notably, *Tetrahymena* are difficult to trap because they move to find nutrients using cilia. To trap *Tetrahymena*, we applied the magnetic cell trapping system schematically illustrated in Fig. 3a (see "Methods"). As previously shown[39], *Tetrahymena* can take up iron oxide nanoparticles, and their movement can then be controlled by an external magnetic field. The presence of iron oxide nanoparticles was clearly visible in the cytoplasm in the *Tetrahymena* (Fig. 3c), whereas the *Tetrahymena* without the particles displayed normal cytoplasm (Fig. 3b). It is worth noting that iron oxide nanoparticles are not known to impair any cellular activity[40]. With the application of an external magnetic field, the ingested iron oxide nanoparticles produced an induced magnetic dipole and were aligned with the direction of the magnetic force, as shown in Fig. 3d. The *Tetrahymena* with the particles can then be attracted and eventually trapped in the presence of the external magnetic field. By aligning the middle of the calorimeter with the interface of the two magnets (Figs. 1e and 3a), we were able to place and trap individual *Tetrahymena* cells sitting atop of the thermopile junction in the middle of the microfluidic channel.

As a single *Tetrahymena* was trapped, nutrients and oxygen had to be supplied to maintain their normal metabolism. Notably, without a continuous supply of oxygen, the amount of oxygen contained in the microfluidic channel and transported to *Tetrahymena* would have been limited because of the small volume and cross-sectional area. The oxygen consumption of *Tetrahymena* is known to be 0.1–1.0 nLO$_2$ cell$^{-1}$ h$^{-1}$[34]; however, our analysis indicated that the amount of oxygen in the channel was not sufficient for a single *Tetrahymena* (Supplementary Fig. 5). Furthermore, previous studies have shown that starved protozoa

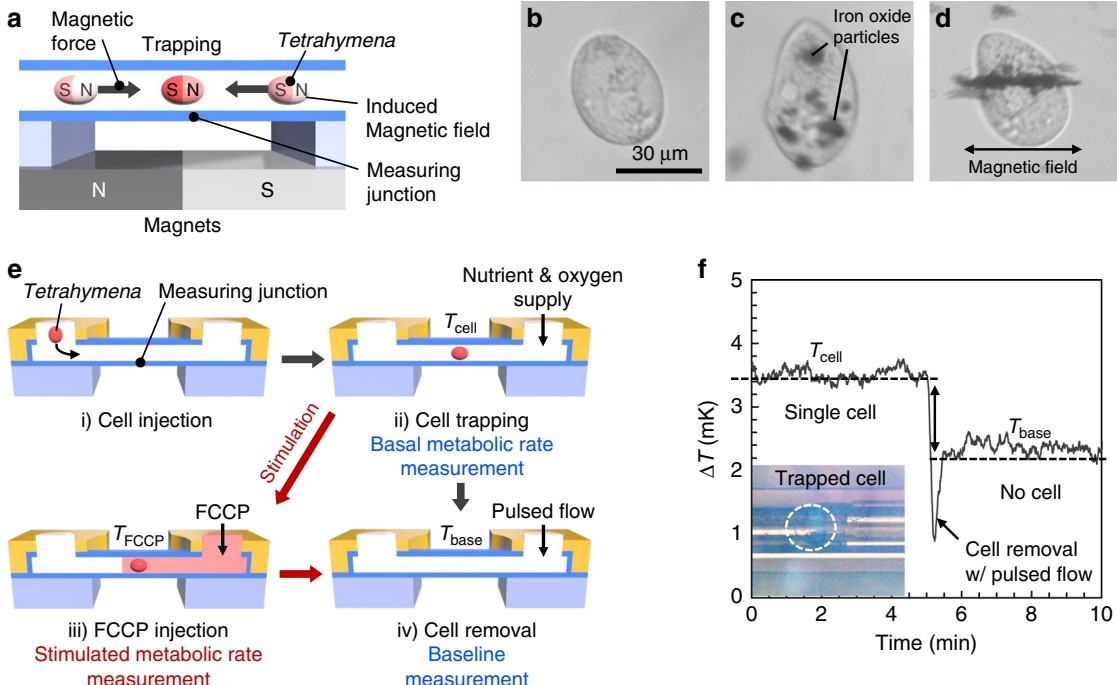

**Fig. 3 Magnetic cell trapping and single-cell measurement procedure. a** Illustration of magnetic cell trapping. The external magnetic field generated by two magnets trapped a single cell, which is internalized with iron oxide nanoparticles at the interface of the magnets, where the highest magnetic field occurred. **b** *Tetrahymena* before internalizing iron oxide nanoparticles. **c** *Tetrahymena* after internalizing iron oxide nanoparticles. **d** *Tetrahymena* with the iron oxide nanoparticles under external magnetic field. More than five ($n > 5$) similar images of cells with nanoparticles were taken. **e** Procedure of single-cell metabolic rate measurement: (i) injection of a single *Tetrahymena* through the microfluidic channel, (ii) trapping of the cell in the middle of the suspended tube by the magnetic field and measurement of the temperature with the cell ($T_{cell}$), (iii) FCCP injection through the microfluidic channel to stimulate cell metabolism and measurement of the temperature with the stimulation ($T_{FCCP}$), and (iv) cell removal with a pulse flow to measure baseline temperature ($T_{base}$). **f** Representative result of the temperature signal during the single-cell measurement. Solid line represents the raw data of the temperature signal. Dashed lines represent the average values before and after cell removal ($T_{cell}$ and $T_{base}$, respectively) The inset photograph shows the *Tetrahymena* trapped by the magnetic field during the measurement. Source data for 3f are provided as a Source Data file.

have a smaller metabolic rate than normal protozoa[29,34]. To supply sufficient nutrients and oxygen, we applied a 0.1 mm s⁻¹ flow of the growth medium to the trapped *Tetrahymena*. The oxygen supply by the flow was estimated to be ~3.4 nL O₂ h⁻¹ at this flow rate, which is a few times larger than the estimated oxygen consumption of single *Tetrahymena*. This low flow rate was insufficient to overcome the magnetic force trapping *Tetrahymena*; therefore, the trapped *Tetrahymena* were not moved by the flow during the measurement (Supplementary Movie 1 (cell trapping)). In addition, the flow did not have a major effect on the calorimeter sensitivity. We determined the thermal conductance of the calorimeter with the flow using the same method described in Fig. 2b to verify the amount of possible advection heat transfer by the flow. The result indicated that the advection heat transfer was almost negligible; thermal conductance with the flow was only increased by ~4% compared with the no-flow condition (from 2.51 to 2.62 μW K⁻¹, Supplementary Fig. 7a). Furthermore, this low-speed constant flow was shown to have negligible effect on the temperature stability in the fluidic channel (Supplementary Fig. 7c). This was mainly achieved by incorporating a 2 cm-long serpentine microfluidic channel on the temperature-controlled Si substrate to preheat the fluid and stabilize its temperature before it enters the suspended channel (Supplementary Fig. 1c). The temperature signal from the suspended calorimeter chamber before and after injecting the fluid (at constant speed of 0.1 mm s⁻¹, corresponding to 0.175 nL s⁻¹ flow rate) was the same, expect for the small temporary drop for a duration of about 1 min after the injection (Supplementary Fig. 4e). In contrast, pulsed flow with a high flow rate can cause temporary temperature

reduction. For example, pulsed fluid injection with ~ 10 nL s⁻¹ flow rate caused ~ 0.8 mK temporary temperature drop (Supplementary Fig. 4c), because the serpentine channel is not long enough to allow a sufficient residence time for high-rate pulsed flow to reach thermal equilibrium (about 3.5 s compared with 200 s for 0.175 nL s⁻¹ flow). We observed this instant temperature drop due to the pulsed fluid injection when we removed the cells from the calorimeter for baseline measurement (Fig. 3f). This temperature drop was recovered in <1 min as soon as we restored the flow rate to the baseline level of 0.1 mm s⁻¹.

The procedure and a representative result of the metabolic rate measurement of single cells are displayed in Fig. 3e, f, respectively. First, a single *Tetrahymena* was injected through the microfluidic channel and trapped on the measuring junctions by a magnetic field. Then, the thermopile recorded the temperature signal of the middle of the microfluidic channel with the trapped cell ($T_{cell}$) while 0.1 mm s⁻¹ flow of growth medium supplied a sufficient amount of nutrients and oxygen. After the measurement, the cell was removed from the calorimeter by a pulsed flow, which was sufficiently strong to overcome the magnetic force. Then, the temperature signal without the cell ($T_{base}$) was recorded as a baseline. In Fig. 3f, the sudden temperature change at 5 min was caused by the pulsed flow for cell removal. This measurement process is shown in Supplementary Movie 2 (cell injection) and Movie 3 (cell removal).

**Single-cell metabolic rate measurements.** We successfully measured the metabolic rate of single cells. The temperature signal

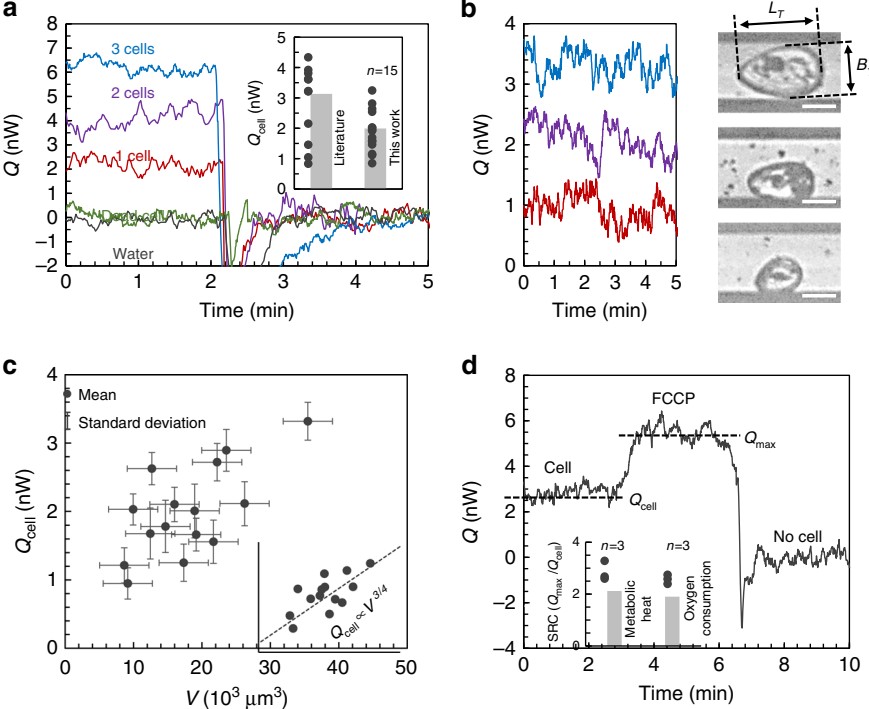

**Fig. 4 Metabolic rate measurement results. a** Basal metabolic rate ($Q_{cell}$) of one (red line), two (purple line), three (blue line), and dead cells (green line) as well as the heat signal from water injection without cells (black line). A clear increase of the heat signal was observed with increasing number of cells, whereas no heat signal was detected for the dead cell or water injection without cells. The inset graph compares the *Tetrahymena* metabolic rate from the literature (3.1 ± 1.4 nW, where the ±error represents standard deviation)[29-31] and this work (2.0 ± 0.7 nW)), where $n = 15$ independent single cell measurements were performed. **b** Single-cell metabolic rate for various sizes of *Tetrahymena*. The large cell ($35 \times 10^3$ μm³, blue line), medium cell ($16 \times 10^3$ μm³, purple line), and small cell ($9 \times 10^3$ μm³, red line) exhibited metabolic rates of 3.3, 2.1, and 0.9 nW, respectively. The scale bars in the images all represent 30 μm. $n = 15$ independent single cell images were taken, corresponding to the 15 calorimetry of single cells. **c** Summary of single-cell metabolic rate as a function of cell size ($V$). Black dots represent individual mean value of $Q_{cell}$ collected for 5 min from $n = 15$ independent single cell measurements with different cells. The x-axis error bars represent standard deviation of heat signals measured every 0.5 s for 5 min ($n = 600$) while the y-axis error bars are based on possible standard deviation errors of cell size measurement (see "Methods" section for details). The inset presents the results on the logarithmic scale. The dotted line in the inset represents a three-quarter power allometric scaling relationship ($Q_{cell} \propto V^{3/4}$). **d** Representative metabolic heat measurement result with FCCP stimulation. The cell metabolic rate was increased upon dosing FCCP to the cell at 3 min among $n = 3$ independent experiments with different cells. The inset shows the SRC (ratio of maximum metabolic rate, $Q_{max}$, to $Q_{cell}$) measured by our single-cell calorimeter and a conventional oxygen electrode chamber on a bulk volume of *Tetrahymena*. Source data for 4**a**-**d** are provided as a Source Data file.

measured from the thermopile when a cell was trapped in the calorimeter ($T_{cell}$) was higher than the baseline temperature after removing the cell ($T_{base}$). This temperature rise ($T_{cell} - T_{base}$) indicates the heat production by cellular metabolic activity of a single *Tetrahymena* and can be converted into metabolic rate ($Q_{cell} = G \times (T_{cell} - T_{base})$). As displayed in the inset graph of Fig. 4a, the average basal metabolic rate of *Tetrahymena* was 2.0 ± 0.7 nW cell⁻¹. This result falls within the range of published values (from 1 to 6 nW cell⁻¹ [29-31]); however, our data provide a much more precise value at the individual cell level. As a control experiment, we also measured the heat generation rate from dead cells as well as flowing water without cells, which indicated no temperature rise from the baseline (Fig. 4a). In addition, the temperature rise and heat generation of two and three cells were also measured, which clearly showed an increase of the heat generation from ~2 nW for one cell to ~4 nW for two cells and finally to ~6 nW for three cells. These experiments confirmed that the detected temperature rise originated from the cell metabolic heat production. With these measurements, we further quantified the cell metabolic heat rate as a function of individual cell size (Fig. 4b); the results are summarized in Fig. 4c (with all the raw data for each measurement presented in Supplementary Fig. 8).

**Real-time monitoring of metabolic rate stimulation.** We further demonstrated stimulation of the metabolic rate by injecting the uncoupling agent FCCP. The measurement procedure was the same as that shown in Fig. 3e, with the added step of injecting FCCP at a constant flow rate of 0.1 mm s⁻¹ into the microfluidic channel after the initial temperature of the microchannel with cell was stabilized. After trapping a single *Tetrahymena*, the heat signal from the cell indicated a typical basal metabolic rate. When FCCP was injected into the calorimeter, the heat generation from the cell gradually increased and eventually reached a maximum ($Q_{max}$) in 1–2 min (Fig. 4d). This stimulated heat signal was collected for 3 min, and then the baseline was measured again after removing the cell with the pulsed flow. The basal metabolic heat generation of *Tetrahymena* increased by a factor of 2.1 ± 0.3 with FCCP stimulation. We also measured the change in the oxygen consumption rate of a bulk volume of *Tetrahymena* upon FCCP stimulation using a Hansatech Oxytherm apparatus[41]. The basal and stimulated oxygen consumption rate of *Tetrahymena* were measured to be 0.20 ± 0.01 and 0.39 ± 0.04 nLO₂ h⁻¹, respectively. These measurements indicated that the oxygen consumption rate increased by a factor of 1.9 ± 0.1 with the FCCP stimulation (inset of Fig. 4d), which is consistent with our calorimeter results.

## Discussion

The basal metabolic heat is mainly produced by mitochondrial activity such as oxidative phosphorylation, generating a proton gradient across the inner membrane of mitochondria, and synthesis of ATP using the proton gradient[42]. Therefore, the mitochondria content in cells largely affects the basal metabolic heat generation[31]. As this mitochondria content increases with the growth of cytoplasm[38], eventually the metabolic rate is a function of cell size during their growth phases (G1–G2 phases before division)[43]. In fact, this allometric scaling relationship between the metabolic rate and body size have been extensively studied in a wide range of organisms from the smallest unicellular microorganism[34] to the largest mammal[44] and is known to follow a three-quarter power relationship:

$$B = B_0 V^{3/4} \tag{2}$$

where, $B$ is the basal metabolic rate, $B_0$ is the normalization coefficient, and $V$ is the body size. Previous studies of the cellular allometric scaling relationship have focused on the relationship between groups of cells[34,44] because conventional metabolic rate measurement systems can only measure the average value of a bulk amount of cells. The metabolic rate change of individual cells during growth have yet to be investigated because of the lack of single-cell measurement system and has only been assumed by studying synchronous cultures[31] or using optical method[38].

In our single-cell measurement, we observed that the metabolic rate of *Tetrahymena* was correlated to the cell size (Fig. 4c) with a correlation of $r^2 = 0.42$. Our results reveal the clear increase of the metabolic rate as a function of cell size. As indicated by the representative results presented in Fig. 4b, a small cell ($9 \times 10^3 \, \mu m^3$) generated metabolic heat of 0.9 nW, whereas a large cell ($35 \times 10^3 \, \mu m^3$) produced metabolic heat as high as 3.3 nW, with a clear difference in their heat signals. Considering that the cell metabolic rate can also be affected by other factors (e.g., gene, protein expression, etc.), the observed correlation between the metabolic rate and cell size is significant. It is also important to note that this correlation approximately follows the conventional three-quarter power allometric scaling relationship (inset of Fig. 4c), which suggests that the scaling relationship is also applicable in individual cells.

FCCP can uncouple the respiratory chain from ATP synthesis by direct transport of protons across the inner membrane of mitochondria and eventually dissipate the proton gradient[42]. This uncoupled respiration maximizes the mitochondrial metabolic activity, which provides useful information about the cellular metabolism[45]. For instance, a large SRC (ratio of $Q_{max}$ to $Q_{cell}$) of more than 5.0 can be observed in postmitotic cells such as adipocytes, myocytes, or neurons, which means the cells can appropriately respond to an increased ATP demand and withstand periods of stress. In contrast, most proliferating cells exhibit a SRC below 3.0 because of the high basal metabolism required to meet the energy demands of cell division. For *Tetrahymena*, although one study reported a sixfold increase of the oxygen consumption rate with FCCP treatment[35], other studies have reported SRC values of ~1.5–3.0[36,37] under regular growth conditions. In addition, most proliferative unicellular as well as small multicellular organisms have SRC values of <3.0 with FCCP treatment. For instance, the SRC has been reported to be 1.4 for protozoa (e.g., *Trypanosoma cruzi*)[46], 1.5–3.0 for mammalian cells[45], and 1.5–2.0 for worms (*Caenorhabditis elegans*)[27]. The SRC of *Tetrahymena* measured using our calorimeter (SRC = 2.1) was similar to the typical SRC of other proliferative cells and agrees well with our oxygen consumption rate measurement (SRC = 1.9, Fig. 4d). These comparisons suggest that the enhanced heat signal with FCCP injection in the calorimeter was generated by the stimulated metabolic rate of *Tetrahymena*. In addition, the SRC of 1.9–2.1 indicates the *Tetrahymena* used in our demonstration were in a normal proliferative state at the time of the measurement.

In summary, we develop a high-sensitivity microfluidic chip calorimeter and demonstrated calorimetric measurement of heat production by single cells. The design with a one-dimensional suspended microfluidic channel and extensive thermal shielding results in significant reduction of the thermal conductance to as low as $2.51 \, \mu W \, K^{-1}$ and low long-term noise of 80 μK, thus enabling a high calorimetric sensitivity of 0.2 nW for single-cell metabolic rate measurement. We measure the basal metabolic heat of single *Tetrahymena* and reveal a clear correlation between the metabolic rate and size of individual cells. Finally, we stimulate metabolism of *Tetrahymena* by dosing the mitochondrial uncoupling agent FCCP through the microfluidic channel and achieve excellent agreement between our heat-generation and oxygen-consumption measurements. These demonstrations suggest that our sub-nanowatt microfluidic calorimetry could be utilized for noninvasive, real-time study of single-cell behaviors, such as cell proliferation[5] and embryonic development[6] as well as for cell cytometry and phenotyping[8,10].

## Methods

**Device fabrication.** The fabrication process is illustrated in Supplementary Fig. 1a. First, a 2-in. Si wafer surface was treated with A-174 adhesion promoter to improve the adhesion between the Si and parylene. A 3-μm-thick first parylene layer was deposited on the Si wafer using a PDS 2010 Parylene Coater. Next, 60-nm-thick Pt and 500-nm-thick Bi layers were sequentially sputtered on the first parylene layer and patterned by photolithography and a lift-off process to form the thermopile. After that, the second parylene layer (3-μm thick) was deposited on the metal layers to passivate the thermopile. Then, 35-μm-thick photoresist (PR) was patterned by photolithography to form a template of the microfluidic structure, which was later removed. The third parylene layer (6-μm thickness) was deposited on the PR template to fabricate the microfluidic channel. Subsequently, the parylene layers where the inlet/outlet holes and the Si etching window would be formed were etched by reactive-ion etching. A 60-μm thick SU-8 layer was then patterned to protect the parylene microfluidic channels. The device was then immersed in propylene glycol monomethyl ether acetate to dissolve the PR microfluidic template structure. Finally, the suspended structure was released by etching the Si underneath the microfluidic tubes using XeF₂ gas.

**Device characterization.** The thermal conductance and sensitivity of the calorimeter were measured using a Pt thin film heater embedded in the suspended microfluidic tubes. First, the temperature coefficient of resistance (TCR) of the Pt film was measured by changing the sample stage temperature and monitoring the resultant electrical resistance change. The measured TCR was $0.00189 \, K^{-1}$. Then, heating power was applied to the Pt heater and gradually increased while the voltage generation of the Bi–Pt thermopile and resistance change of the Pt heater were monitored at a constant stage temperature. The resistance change of the Pt heater was converted to the average temperature change along the suspended tube using the TCR. The overall thermal conductance of the calorimeter ($G$) and Seebeck coefficient of the Bi–Pt thermopile ($S$) were calculated by COMSOL simulation using the applied power to the Pt heater and resultant average temperature change. In addition, the absorbed heat by the measuring junctions ($Q_{abs}$) during the Joule heating by the Pt heater, which heated the entire suspended tube, was estimated by the COMSOL simulation. The thermal conductance was calculated based on $Q_{abs}$ and the corresponding $\Delta T_{mea}$ change ($G = Q_{abs}/\Delta T_{mea}$).

The material parameters of parylene used in the COMSOL simulation were experimentally measured; the in-plane thermal conductivity and emissivity were measured to be $0.18 \, W \, m^{-1} \, K^{-1}$ and 0.21, respectively (Supplementary Fig. 2). In addition, the thermal conductivity of the Pt thin film used in the simulation was ~ $54 \, W \, m^{-1} \, K^{-1}$, which was estimated from the measured electrical conductivity and the Wiedemann–Franz law ($k = \sigma L T$, where $k$ is the thermal conductivity, $\sigma$ is the electrical conductivity, $L$ is the Lorenz number, and $T$ is the temperature), and the thermal conductivity of Bi thin film was assumed to be ~ $4 \, W \, m^{-1} \, K^{-1}$ based on the literature values for similar films[47].

**Measurement platform.** The measurement platform was composed of three temperature-controlled layers: an outer chamber, a vacuum chamber, and a sample stage. The temperature of the layers was kept constant by temperature controllers (PTC-10, Stanford Research Systems), thin film heaters, and a thermoelectric device, as illustrated in Fig. 1d, e. The outer chamber and vacuum chamber were insulated by electrically neutral balsa wood to prevent electrostatic build-up, and

the sample stage was insulated by a high-vacuum environment. Thermal radiation from the immediate surroundings was blocked by a microfluidic clamp, which covered the calorimeter as a radiation shield with only a small viewport to see the microfluidic channel. The microscope illumination was provided from a LED light source with a color temperature of 5000 K, which has no infrared component. To avoid excessive heating by the illumination (could be as high as ~100 mK in temperature rise and a few hundred μK in temperature noise if a strong illumination was used), the LED light intensity was adjusted to the minimal level needed for the cell observation in the microchannels, which resulted in no noticeable temperature noise (Supplementary Fig. 7c). In addition, we used an aluminum cover with a small viewport size as a radiation shield. The viewport on the clamp was further covered by two separated pieces of IR-absorbing glass to prevent the transmission of the thermal radiation from the light and the environment, as shown in Supplementary Fig. 4f. These thermal radiation shields substantially reduced the temperature drift in the microfluidic channel from ~5 mK to ~80 μK (Fig. 2a, b, Supplementary Fig. 4g, h).

For accurate and leakage-free fluid control, the inlet/outlet of microfluidic channel in the calorimeter were connected to an external fluid control system by a custom-made microfluidic clamp. Precisely designed microfluidic fittings were screwed on the microfluidic clamp and provided a vacuum-sealed and small dead-volume fluid connection between the microfluidic channel and the tubes. The small dead volume of the platform enabled accurate fluid control as well as efficient cell and stimuli injection, and vacuum-tight sealing prevented temperature agitation by fluid evaporation. The small inner size of the tubes (50 μm in diameter) also minimized the dead volume of the fluid system and the inlet/outlet tubes passed through the lead of the vacuum chamber with a vacuum-sealed feedthrough.

A syringe pump (Pump 11 Pico Plus, Harvard Apparatus, MA, USA) and a 100 μL syringe (Hamilton Company, NV, USA) were used to pump the fluid into the microchannel. Considering the cross-section area of the microfluidic tube (35 × 50 μm$^2$) and the inner diameter of the syringe (1.4 mm), a volumetric flow rate of 0.175 nL s$^{-1}$ was applied to yield fluid velocity of 0.1 mm s$^{-1}$ in the microfluidic tube, in order to continuously supply the nutrient and oxygen to the cells.

**Cell preparation**. *Tetrahymena thermophila* SB210 cells (Tetrahymena Stock Center, Cornell University) were axenically cultured to middle log phase in a liquid medium containing 0.25% yeast extract, 0.25% proteose peptone, 0.5% D-Glucose, and 1 mM FeCl$_3$. Then, 0.1 vol% iron oxide nanoparticles (40 nm in diameter) were added to the culture medium with *Tetrahymena* and gently mixed. The resultant suspension was allowed to stand for 1 h to ensure that the *Tetrahymena* took up the nanoparticles. The magnetic trapping effect on cells was verified by placing a droplet of the cells on a glass slide and applying a magnetic field before injecting the cells in the calorimeter.

A fixation buffer for the dead cell measurement was prepared by mixing 1 mL of 37% formaldehyde solution, 1 mL of phosphate buffer solution (PBS), and 8 mL of water. *Tetrahymena* was centrifuged to aspirate the growth medium, and the fixation buffer was added to kill the cells. After incubation at room temperature for 10 min, the cells were washed with PBS and stored in 70% ethanol solution.

**Magnetic cell trapping**. We used two NdFeB permanent magnets embedded in the sample stage to generate the magnetic field for cell trapping (Fig. 1e); the first magnet was placed with its south pole upwards, and the second magnet was placed with its north pole upward next to the first magnet. The iron oxide nanoparticles were mixed with the culture media, and *Tetrahymena* took up the particles before measurement. The cells containing the iron oxide nanoparticles were trapped at the interface of the two magnets where the highest magnetic field occurred (Supplementary Movie 1 (cell trapping)).

**Cell volume measurement**. The volume of *Tetrahymena* (V) was calculated based on the morphometric analysis with the assumption that the cells had a prolate spheroid shape[48]:

$$V = \frac{\pi}{6} L_T B_T^2, \tag{3}$$

where $L_T$ is the length and $B_T$ is the breadth of *Tetrahymena*. The standard deviation of this optical estimation compared with electrical measurement (Coulter counter)[48] is ~1800–5500 μm$^3$.

**Oxygen consumption rate measurement**. The oxygen consumption rate was measured using a Hansatech Oxytherm (Hansatech, King's Lynn, UK). The instrument was calibrated at 28 °C per the manufacturer's instructions. *Tetrahymena* were grown and assayed in normal culture media as detailed above. Then, 0.5 mL of cell culture with concentration of 280,000 *Tetrahymena* per mL was added to the chamber under constant magnetic stirring. After the basal oxygen consumption rate was measured, successive 0.2-μL additions of 0.2 mM FCCP were added until a maximal uncoupler-stimulated respiration rate was achieved.

**Reporting summary**. Further information on research design is available in the Nature Research Reporting Summary linked to this article.

## Data availability

Source data for Figs. 2a, b, 3f, 4a–d and Supplementary Figs. 4c, d, e, g, h, 7a, 7c, 8 have been provided as a Source Data file. All other data supporting the findings of this study are available from the corresponding author upon reasonable request. Source data are provided with this paper.

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

## Acknowledgements

R.C. acknowledges the startup fund from UCSD and National Science Foundation (CBET-1336428 and DMR-1508420) for the support of this work. C.M. acknowledges the support from National Science Foundation (CAREER Award CBET-1454425).

## Author contributions

R.C. and C.M. conceived the project. S.H. and E.D. designed and fabricated the devices and made the setup. S.H. did the calorimetry experiments. C.G., A.M., C.M. contributed to cell culturing and oxygen consumption measurements. S.H., C.G., C.M., and R.C. interpreted the results. S.H. and R.C. wrote the manuscript with substantial contributions from E.D., C.G., and C.M. All the authors discussed the data and edited the manuscript.

## Competing interests

The authors declare no competing interests.
