## [Peer Review File · Nature Communications]

Reviewers' Comments:

Reviewer #1:

Remarks to the Author:

The high thermal sensitivity of sub-nanowatt is very interesting, and the characteristics of the device are well described basing on the experiments and theories. My comments are as follows.

* What is your original and novel technique in the paper?

The focusing points of the presented calorimeter are (1) Thermal insulation by vacuum surrounding the microchannel, (2) trapping individual cells in the microfluidic channel, (3) supplying sufficient nutrients and oxygen to the trapped cell; However, the device is composed of a combination of already-reported techniques (e.g. (1): ref [16], (2): ref [39]); therefore, it does not have a clear novelty to readers from the technical viewpoint. (3) seems to be a problem of only the flow rate, not technical matters.

*How do you define "non-invasive methods"?

Author introduced taking up of magnetic particles into cells as a non-invasive method; however, I do not think it is "exactly" non-invasive methods because the cell takes up the particles which naturally are not contained in the cell even if they are not toxic against cells.

Reviewer #2:

Remarks to the Author:

The paper describes a chip calorimeter with a sensitivity, based on minimum detectable power (nW), that is approximately an order of magnitude better than previously reported chip calorimeters. The architecture of the calorimeter is similar in some respects to that reported by Lee et al (in particular the vacuum isolation of the measurement chamber), but there are some significant improvements and differences to differentiate it from prior calorimeters.

The ability to measure single cell metabolism (albeit on highly energetic cells), should be of interest not only to those working in the nanocalorimetry/chip calorimetry field, but also potentially to cell biologists.

Overall, the paper is well written and the authors do not try to oversell their claims of the ability of the calorimeter to measure various type of cells. I noted a few areas where the description of the methods and the specific details of the calorimeter measurement platform were unclear or missing:

Was illumination constant during the measurements? What type of illumination was used (halogen, LED, something else)? If halogen-based, was the IR component filtered? The manuscript only describes "microscope illumination", but if the illumination source is constantly turned on, the stability of the illumination looks to be critical to long-term baseline stability (shown in Figure 2 and supplementary material). This is particularly highlighted in supplementary figure 4 – turning on illumination for ~10s seems to produce temperature changes greater than 1 mK. If illumination is constant during the measurements, there would likely be temperature fluctuations of at least 30 μ K due to illumination.

What type of pump was used to supply the liquid flow (syringe, pressure, other)? Again, some types of pumps have pulsing and/or instability in their flow that would be a source of temperature fluctuation, particularly in light of the temperature changes shown in Supplementary Figure 4 for 10 nl fluid injection which suggest the incoming fluid is cooler than the measurement chamber. If the flow of nutrients/oxygen is constant, a highly stable flow rate is essential to achieve the stability shown in Figure 2.

Was the FCCP injected as a pulse or as a constant flow? It is unclear as written.

Additional clarification in the abstract: the statement "...challenge of stabilizing the cell temperature at the sub-mK level.." is unclear. Is this the temperature of the calorimeter cell or the biological cell? If it is the biological cell, wouldn't there be processes that might cause fluctuations on the mK level? It seems to me that the goal is to have a calorimeter measurement with high stability (sub mK level) to allow measurements of cell fluctuations at the sub-mK to mK level.

Overall, if the omissions noted above are addressed, I think that the work is of sufficient novelty and interest to be accepted for publication.

Response to the reviewers' comments

We thank the recommendation and constructive comments from the reviewers. Herein we list our point-by-point responses to the reviewers' comments.

Color codes used in this response letter:

Blue Italic: original review comments;

Black: our responses;

Red Italic: revisions made in the manuscript.

Reviewer #1:

“The high thermal sensitivity of sub-nanowatt is very interesting, and the characteristics of the device are well described basing on the experiments and theories. My comments are as follows.”

Response: We greatly appreciate the reviewer's careful examination of our manuscript and for the favorable recommendation.

Reviewer #1 – Q1:

“What is your original and novel technique in the paper?”

The focusing points of the presented calorimeter are (1) Thermal insulation by vacuum surrounding the microchannel, (2) trapping individual cells in the microfluidic channel, (3) supplying sufficient nutrients and oxygen to the trapped cell; However, the device is composed of a combination of already-reported techniques (e.g. (1): ref [16], (2): ref [39]); therefore, it does not have a clear novelty to readers from the technical viewpoint. (3) seems to be a problem of only the flow rate, not technical matters.”

Response:

We appreciate the reviewer for the valuable comments and would like to elaborate the novelty of this work. In our opinion, the most important result in this work is the first report of *sub-nanowatt* calorimetry (~ 0.2 nW) based on a *microfluidic* platform, which represents more than 10-fold enhancement over the state of the art (4 nW in Ref. 16, by Prof. Roukes group at Caltech in a PNAS paper in 2009) as well as the *first demonstration of single-cell calorimetry* (especially with a microfluidic system). These results were enabled by implementing a combination of a series of techniques. While many of these techniques were indeed already reported as the reviewer pointed out, we have significantly improved some of the individual techniques and, more importantly, we are also the first one to integrated all these techniques into a single platform to achieve the unprecedentedly calorimetry resolution and flow stability that enabled the single cell measurement.

More specifically, the innovations include:

- 1) Realization of one-dimensional (1D) microfluidic tube structure with optimized geometry and material selection (e.g., ***Supplementary Figure 3***). Although both ref[16] and our calorimeter eliminated thermal convection via vacuum surrounding, our 1D calorimeter has about 7x smaller thermal conductance ($2.5 \mu\text{W K}^{-1}$) compared to the 2D membrane-based calorimeter shown in ref[16] thanks to the large reduction of surface and cross-section area in our design.
- 2) Minimization of the influence of external thermal agitation with a carefully designed temperature stabilization system including viewport/radiation shields (newly added

Supplementary Figure 4f). We also reduced the thermal noise and temperature drift from environment and the microscope illumination. As shown in the newly added *Supplementary Figure 4g*, the temperature drift of the calorimeter largely followed the ambient temperature change despite the use of multiple layers of temperature control (*Figure 1d*). We found this large temperature drift is mainly caused by the radiation heat transfer through the viewport for the microscope, which has not been considered in previous work. We used multiple radiation shielding and blocking mechanisms (newly added *Supplementary Figure 4f*) to minimize the radiation heat transfer through the view port, which resulted in the elimination of long-term temperature drift as well as dramatical reduction of short-term thermal noise (newly added *Supplementary Figure 4h*).

- 3) Single cell trapping is a common technique used in the cytometry field, and we understand that there are various methods to trap cells. However, most of the cell trapping methods are difficult to be applied to the present calorimetry scheme because they could affect thermal measurement for different reasons. The novelty of our work lies in the cell control without the losing the calorimetry resolution. For instance, common cell trapping methods using fluid pressure difference (e.g., funnel cell trapping) would occupy a large space, resulting in higher thermal conductance. The electromagnet used in ref [39] to control *Tetrahymena* could cause large electric noise in the thermometry system (which has a sensitivity of nanovolt and micro-Kelvin) and is too bulky for our temperature control setup. We combined the small rare-earth permanent magnets (NdFeB) in the temperature control stage and the fluid flow control to trap/release single cells, which is also a new strategy.
- 4) Supplying nutrient into the microfluidic device in a vacuum chamber without thermal agitation is also a big challenge in this work. We optimized the parylene wall thickness of the suspended microfluidic tube to prevent permeation/leakage of water and oxygen (if the parylene layer is too thin) and to avoid large parasitic heat conduction loss along the tube (if the parylene layer is too thick). When the parylene layer was very thin, we have observed that water vapor and oxygen could permeate from the tube to the vacuum environment, resulting in large temperature fluctuation (by over 1 deg. C) and loss of oxygen in the fluid. Preventing thermal agitation during the nutrient supply was also a challenging issue. Since the nutrient temperature was lower than calorimeter temperature before injection, nutrient supply without proper temperature control caused temperature drop of calorimeter. We incorporated a 2 cm-long serpentine microfluidic channel on the temperature-controlled Si substrate (newly added *Supplementary Figure 1c*) to preheat the fluid and stabilize its temperature before it enters the suspended channel.

We also added more details regarding the temperature stability of the calorimeter chamber when injecting fluids in the revised manuscript (**page 10, line 12**)

“Furthermore, this low-speed constant flow was shown to have negligible effect on the temperature

stability in the fluidic channel. This was mainly achieved by incorporating a 2 cm-long serpentine microfluidic channel on the temperature-controlled Si substrate (*Supplementary Figure 1c*) to preheat the fluid and stabilize its temperature before it enters the suspended channel. The temperature signal from the suspended calorimeter chamber before and after injecting the fluid (at constant speed of 0.1 mm s^{-1} , corresponding to 0.15 nL s^{-1} flow rate) was the same, except for the a small temporary drop ($< 1 \text{ min}$) right after the injection (*Supplementary Figure 4e*). In contrast, pulsed flow with a high flow rate can cause temporary temperature reduction. For example, pulsed fluid injection with $\sim 10 \text{ nL s}^{-1}$ flow rate caused $\sim 0.8 \text{ mK}$ temporary temperature drop (*Supplementary Figure 4c*), because the serpentine channel cannot respond fast enough to stabilize the high-rate pulsed flow. We observed this instant temperature drop due to the pulsed fluid injection when we removed the cells from the calorimeter for baseline measurement (*Figure 3f*). This temperature drop was recovered in less than 1 min as soon as we restored the flow rate to the baseline level of 0.1 mm s^{-1} .”

We have added following supplementary figure.

Supplementary Figure 1c. Serpentine microfluidic channel for nutrient temperature stabilization.

Also, we added experimental data showing that flow rate of 0.1 mm s^{-1} did not cause temperature change in *Supplementary Figure 4e*.

Supplementary Figure 4e. Temperature stability with no flow and after applying continuous flow at 0.1 mm s^{-1} (corresponding to 0.15 nL s^{-1} flow rate from the syringe pump)

We have now specified the type of illumination and the associated details in the revised manuscript (page 16, line 1):

“The microscope illumination was provided from a LED light source with a color temperature of 5000 K, which has no infrared component. To avoid excessive heating by the illumination (could be as high as ~ 100 mK in temperature rise and a few hundred μK in temperature noise if a strong illumination was used), the LED light intensity was adjusted to the minimal level needed for the cell observation in the microchannels., which resulted in no noticeable temperature noise. In addition, we used an aluminum cover with a small viewport size as a radiation shield. The viewport on the clamp was further covered by two separated pieces of IR-absorbing glass to prevent the transmission of the thermal radiation from the light and the environment, as shown in *Supplementary Figure 4f*. These thermal radiation shields substantially reduced the temperature drift in the microfluidic channel from ~ 5 mK to ~ 80 μK (*Figure 2a and 2b, Supplementary Figure 4g and 4h*).”

We have also added the following supplementary figure to more clearly show the radiation shields used in the setup to mitigate the noise caused by the illumination.

Supplementary Figure 4. Data acquisition and stability features of the calorimetry. **a**, Photograph and circuit diagram of operational amplifier and low-pass filter. **b**, Time constant of the calorimeter. **c**, Baseline stability with the series of fluid injections with high pulsed flow rate (~ 10 nL s^{-1}). **d**, Baseline stability with the series of light irradiation. **e**, Temperature stability with no flow and after applying continuous flow at 0.1 mm s^{-1} (corresponding to 0.15 nL s^{-1} flow rate from the syringe

pump) **f**, Radiation shields to protect calorimeter from ambient thermal radiation. **g**, Temperature stability without thermal radiation shields. **h**, Temperature stability with thermal radiation shields.

Reviewer #1 – Q2:

“How do you define “non-invasive methods”?”

Author introduced taking up of magnetic particles into cells as a non-invasive method; however, I do not think it is “exactly” non-invasive methods because the cell takes up the particles which naturally are not contained in the cell even if they are not toxic against cells.”

Response:

We appreciate the reviewer for the valuable comments. The “non-invasive method” in the manuscript means we can observe the living cells in real time (e.g., does not need to kill cells), whereas most of the chemical cytometry methods require cell lysis to analyze the metabolites. As the reviewer correctly pointed out, although the iron oxide particles do not seem to cause harmful effect, it may not represent the actual condition where they live in nature. The exact role of the iron oxide nanoparticles at both cellular and molecular levels is perhaps a very complex problem and is beyond the scope of this study. Nevertheless, in this work, we can reasonably speculate that the energetics of the *Tetrahymena* cells did not seem to be altered in the presence of the nanoparticles, as evidenced by the measured metabolic rate that is consistent with the literature values. Qualitatively, in the absence of the magnetic field, the cell movement after the nanoparticle loading also appeared to be similar to that of pristine cells. Since the main focus of this work is to demonstrate single cell calorimetry of trapped single cells, we believe the use of the iron oxide nanoparticles does not change the validity of our conclusion. In addition, there are other methods that can be used to trap single cells, such as geometrical trapping, which could be suitable for less mobile but similarly energetic cells, such as brown adipocytes (*Tetrahymena* cells are especially mobile because of their cilia).

We added the following sentence to more clearly specify the meaning of “non-invasive method” (page 3, line 13):

“Calorimetry is a powerful tool to investigate the metabolic activities because it is a non-invasive (i.e., without the need of cell lysis to extract and analyze the metabolites) and label-free technique, enabling monitoring of the metabolic behavior of living cells in real time without altering the nature of the microorganisms”

Reviewer #2:

“The paper describes a chip calorimeter with a sensitivity, based on minimum detectable power (nW), that is approximately an order of magnitude better than previously reported chip calorimeters. The architecture of the calorimeter is similar in some respects to that reported by Lee et al (in particular the vacuum isolation of the measurement chamber), but there are some significant improvements and differences to differentiate it from prior calorimeters.

The ability to measure single cell metabolism (albeit on highly energetic cells), should be of interest not only to those working in the nanocalorimetry/chip calorimetry field, but also potentially to cell biologists.

Overall, the paper is well written and the authors do not try to oversell their claims of the ability of the calorimeter to measure various type of cells.”

We are very grateful for the reviewer’s positive comments and the recommendation of our work.

Reviewer #2 – Q1:

I noted a few areas where the description of the methods and the specific details of the calorimeter measurement platform were unclear or missing:

Was illumination constant during the measurements? What type of illumination was used (halogen, LED, something else)? If halogen-based, was the IR component filtered? The manuscript only describes “microscope illumination”, but if the illumination source is constantly turned on, the stability of the illumination looks to be critical to long-term baseline stability (shown in Figure 2 and supplementary material). This is particularly highlighted in supplementary figure 4 – turning on illumination for ~10s seems to produce temperature changes greater than 1 mK. If illumination is constant during the measurements, there would likely be temperature fluctuations of at least 30 μ K due to illumination.

Response:

We appreciate reviewer’s careful reading. The reviewer is exactly right in his/her assessment of the noise that can be induced by the microscope illumination. As the reviewer anticipated, elimination of the thermal noise from the microscope illumination was one of the biggest challenges in this work. To mitigate this issue, we used a LED light source with a color temperature of 5000 K (rather than a halogen lamp) to eliminate the IR component in the illumination. When the light intensity was strong (i.e., normal microscope illumination setting), we did observe an increase of the calorimeter temperature by ~100 mK and associated thermal noise of a few hundred μ K. To address this issue, we used minimal light intensity that was just sufficient for cell observation during the cell measurement to minimize the illumination-induced temperature change. We did not see noticeable noise with the low illumination intensity used in this work. To visualize cells with this low light intensity, we maximized the microscope gain and exposure time. We also used an aluminum cover with a small viewport size as a radiation shield. The viewport on the clamp was further covered by two separated pieces of IR-absorbing glass to prevent the transmission of the thermal radiation from the light and the environment, as shown in the new *Supplementary Figure 4f*. The experiments shown in Figure 2a and 2b were performed with the minimized illumination and the radiation shields, which resulted in a small temperature noise of 80 μ K.

We have now specified the type of illumination and the associated details in the revised manuscript (**page 16, line 1**):

“The microscope illumination was provided from a LED light source with a color temperature of

5000 K, which has no infrared component. To avoid excessive heating by the illumination (could be as high as ~ 100 mK in temperature rise and a few hundred μK in temperature noise if a strong illumination was used), the LED light intensity was adjusted to the minimal level needed for the cell observation in the microchannels., which resulted in no noticeable temperature noise. In addition, we used an aluminum cover with a small viewport size as a radiation shield. The viewport on the clamp was further covered by two separated pieces of IR-absorbing glass to prevent the transmission of the thermal radiation from the light and the environment, as shown in *Supplementary Figure 4f*. These thermal radiation shields substantially reduced the temperature drift in the microfluidic channel from ~ 5 mK to ~ 80 μK (*Figure 2a and 2b, Supplementary Figure 4g and 4h*).”

We have also added the following supplementary figure to more clearly show the radiation shields used in the setup to mitigate the noise caused by the illumination.

Supplementary Figure 4. Data acquisition and stability features of the calorimetry. **a**, Photograph and circuit diagram of operational amplifier and low-pass filter. **b**, Time constant of the calorimeter. **c**, Baseline stability with the series of fluid injections with high pulsed flow rate (~ 10 nL s^{-1}). **d**, Baseline stability with the series of light irradiation. **e**, Temperature stability with no flow and after applying continuous flow at 0.1 mm s^{-1} (corresponding to 0.15 nL s^{-1} flow rate from the syringe pump) **f**, Radiation shields to protect calorimeter from ambient thermal radiation. **g**, Temperature stability without thermal radiation shields. **h**, Temperature stability with thermal radiation shields.

Reviewer #2 – Q2:

What type of pump was used to supply the liquid flow (syringe, pressure, other)? Again, some types

of pumps have pulsing and/or instability in their flow that would be a source of temperature fluctuation, particularly in light of the temperature changes shown in Supplementary Figure 4 for 10 nl fluid injection which suggest the incoming fluid is cooler than the measurement chamber. If the flow of nutrients/oxygen is constant, a highly stable flow rate is essential to achieve the stability shown in Figure 2.

Response:

We used a syringe pump (Pump 11 Pico Plus, Harvard Apparatus, MA, USA) and 100 μL syringes (Hamilton Company, NV, USA). Considering the cross-section area of microfluidic tube ($30\ \mu\text{m} \times 50\ \mu\text{m}$) and inner diameter of syringe (1.4 mm), we applied volumetric flow rate of $0.15\ \text{nL s}^{-1}$ to make fluid velocity of $0.1\ \text{mm s}^{-1}$ in the microfluidic tube. This flow rate can be precisely controlled with the syringe pump. More importantly, as the reviewer speculated, the temperature stability of the calorimeter chamber under the flow condition is a very important and challenging aspect of our design. Our calorimeter is designed to stabilize the temperature of suspended fluidic channel with this low flow rate. This was mainly achieved by incorporating a 2 cm-long serpentine microfluidic channel on the temperature-controlled Si substrate to preheat the fluid and stabilize its temperature before it enters the suspended channel (see newly added **Supplementary Figure 1c**, also shown below). As shown in the newly added **Supplementary Figure 4e** (also shown below), the temperature signal from the suspended calorimeter chamber before and after injecting the fluid (at constant speed of $0.1\ \text{mm s}^{-1}$, corresponding to $0.15\ \text{nL s}^{-1}$ flow rate) was the same, except for the small temporary drop for a duration of about 1 min after the injection. In contrast, pulsed flow with a high flow rate can cause temporary temperature reduction. For example, pulsed fluid injection with $\sim 10\ \text{nL s}^{-1}$ flow rate caused $\sim 0.8\ \text{mK}$ temporary temperature drop (**Supplementary Figure 4c**), because the serpentine channel cannot respond fast enough to stabilize the high-rate pulsed flow. We observed this instant temperature drop due to the pulsed fluid injection when we removed the cells from the calorimeter for baseline measurement (**Figure 3f**). Nevertheless, this temperature drop was recovered in less than 1 min as soon as we restored the flow rate to the baseline level of $0.1\ \text{mm s}^{-1}$ (**Figure 3f**).

We have now specified the type of pump and the associated details in the revised manuscript (**page 16, line 21**):

A syringe pump (Pump 11 Pico Plus, Harvard Apparatus, MA, USA) and a 100 μL syringe (Hamilton Company, NV, USA) were used to pump the fluid into the microchannel. Considering the cross-section area of microfluidic tube ($30 \times 50\ \mu\text{m}^2$) and inner diameter of syringe (1.4 mm), a volumetric flow rate of $0.15\ \text{nL s}^{-1}$ was applied to yield fluid velocity of $0.1\ \text{mm s}^{-1}$ in the microfluidic tube, in order to continuously supply the nutrient and oxygen to the cells.

We also added more details regarding the temperature stability of the calorimeter chamber when injecting fluids in the revised manuscript (**page 9, line 12**):

“Furthermore, this low-speed constant flow was shown to have negligible effect on the temperature stability in the fluidic channel. This was mainly achieved by incorporating a 2 cm-long serpentine microfluidic channel on the temperature-controlled Si substrate to preheat the fluid and stabilize its

temperature before it enters the suspended channel (*Supplementary Figure 1c*). The temperature signal from the suspended calorimeter chamber before and after injecting the fluid (at constant speed of 0.1 mm s^{-1} , corresponding to 0.15 nL s^{-1} flow rate) was the same, except for the small temporary drop for a duration of about 1 min after the injection (*Supplementary Figure 4e*). In contrast, pulsed flow with a high flow rate can cause temporary temperature reduction. For example, pulsed fluid injection with $\sim 10 \text{ nL s}^{-1}$ flow rate caused $\sim 0.8 \text{ mK}$ temporary temperature drop (*Supplementary Figure 4c*), because the serpentine channel cannot respond fast enough to stabilize the high-rate pulsed flow. We observed this instant temperature drop due to the pulsed fluid injection when we removed the cells from the calorimeter for baseline measurement (*Figure 3f*). This temperature drop was recovered in less than 1 min as soon as we restored the flow rate to the baseline level of 0.1 mm s^{-1} .”

We have added following supplementary figures:

Supplementary Figure 1c. Serpentine microfluidic channel for nutrient temperature stabilization.

Also, we added experimental data showing that 0.1 mm s^{-1} flow does not cause temperature change.

Supplementary Figure 4e. Temperature stability with no flow and after applying continuous flow at 0.1 mm s^{-1} (corresponding to 0.15 nL s^{-1} flow rate from the syringe pump)

Reviewer #2 – Q3:

Was the FCCP injected as a pulse or as a constant flow? It is unclear as written.

Response:

FCCP injection was performed with a constant flow rate of 0.1 mm s^{-1} of (the same flow rate for delivering the nutrient and oxygen to the channel) to ensure the cell is trapped in calorimeter chamber.

We added the following in the *Real-time monitoring of metabolic rate stimulation* section to better illustrate this point (**Page 11, Line 6**):

*“The measurement procedure was the same as that shown in Fig. 3e, with the added step of injecting FCCP **at a constant flow rate of 0.1 mm s^{-1}** into the microfluidic channel after the initial temperature of the microchannel with cell was stabilized.”*

Reviewer #2 – Q4:

Additional clarification in the abstract: the statement “...challenge of stabilizing the cell temperature at the sub-mK level..” is unclear. Is this the temperature of the calorimeter cell or the biological cell? If it is the biological cell, wouldn't there be processes that might cause fluctuations on the mK level? It seems to me that the goal is the have a calorimeter measurement with high stability (sub mK level) to allow measurements of cell fluctuations at the sub-mK to mK level.

Response:

We appreciate reviewer's careful reading and apologize for the confusion. "cell" in the abstract means the calorimeter cell. We have changed the word "cell" to "chamber" in the abstract to avoid confusion.

(Line 4 in the "Abstract"): *"However, the metabolic heat of single cells has yet to be measured calorimetrically because of the insufficient sensitivity of state-of-the-art microfluidic calorimeters as well as the challenge of stabilizing the **chamber** temperature at the sub-mK level while supplying nutrients and oxygen to the cells."*

"Overall, if the omissions noted above are addressed, I think that the work is of sufficient novelty and interest to be accepted for publication."

Again, we are very appreciative of both reviewers' careful reading of our manuscript and their constructive comments and strong recommendation. We hope that our responses and the revised manuscript are considered suitable for publication in *Nature Communications*.

Reviewers' Comments:

Reviewer #1:

Remarks to the Author:

The author exactly and correctly answered to my comments, and I do not have additional comments.

Reviewer #2:

Remarks to the Author:

I have reviewed the authors' revised manuscript and their point-by-point responses to the original comments. The revised manuscript addresses the concerns I noted in my original review and feel that it is appropriate for publication.

A minor comment that might clarify one section of added text (on p. 9 of the Word document):

"For example, pulsed fluid injection with ~ 10 nL s⁻¹ flow rate caused ~ 0.8 mK temporary temperature drop (Supplementary Figure 4c), because the serpentine channel cannot respond fast enough to stabilize the high-rate pulsed flow."

I would suggest that it is not the channel that is responding to the flow, but rather something along the lines of "the serpentine channel is not long enough to allow a sufficient residence time for high-rate pulsed flow to reach thermal equilibrium (about 3 seconds compared to 200 seconds for 0.15 nL/s flow.)"

Response to the reviewers' comments

We thank the recommendation and constructive comments from the reviewers. Herein we list our responses to the reviewers' comments.

Reviewer #1 (Remarks to the Author):

"The author exactly and correctly answered to my comments, and I do not have additional comments."

Response: We greatly appreciate the reviewer's careful examination of our manuscript and for the favorable recommendation.

Reviewer #2 (Remarks to the Author):

"I have reviewed the authors' revised manuscript and their point-by-point responses to the original comments. The revised manuscript addresses the concerns I noted in my original review and feel that it is appropriate for publication."

*A minor comment that might clarify one section of added text (on p. 9 of the Word document):
"For example, pulsed fluid injection with ~ 10 nL s⁻¹ flow rate caused ~ 0.8 mK temporary temperature drop (Supplementary Figure 4c), because the serpentine channel cannot respond fast enough to stabilize the high-rate pulsed flow."*

I would suggest that it is not the channel that is responding to the flow, but rather something along the lines of "the serpentine channel is not long enough to allow a sufficient residence time for high-rate pulsed flow to reach thermal equilibrium (about 3 seconds compared to 200 seconds for 0.15 nL/s flow.)"

Response: We greatly appreciate the reviewer's strong recommendation and valuable suggestion. We have now revised the sentence accordingly:

On page 9, we have changed "because the serpentine channel cannot respond fast enough to stabilize the high-rate pulsed flow" to "because the serpentine channel is not long enough to allow a sufficient residence time for high-rate pulsed flow to reach thermal equilibrium (about 3.5 sec compared to 200 sec for 0.175 nL s⁻¹ flow)." (Please note that the flow rate value quoted in the last version, 0.15 nL/s, is now corrected to 0.175 nL/s, due to the use of the more exact channel cross sectional area).

Again, we are very appreciative of both reviewers' careful reading of our manuscript and their strong recommendation. We hope that our responses and the revised manuscript are considered suitable for publication in *Nature Communications*.